# The mammalian tRNA ligase complex mediates splicing of *XBP1* mRNA and controls antibody secretion in plasma cells

Jennifer Jurkin[1,†], Theresa Henkel[1,†], Anne Færch Nielsen[2], Martina Minnich[3], Johannes Popow[4], Therese Kaufmann[1], Katrin Heindl[5], Thomas Hoffmann[3], Meinrad Busslinger[3] & Javier Martinez[1,*]

## Abstract

The unfolded protein response (UPR) is a conserved stress-signaling pathway activated after accumulation of unfolded proteins within the endoplasmic reticulum (ER). Active UPR signaling leads to unconventional, enzymatic splicing of *XBP1* mRNA enabling expression of the transcription factor XBP1s to control ER homeostasis. While IRE1 has been identified as the endoribonuclease required for cleavage of this mRNA, the corresponding ligase in mammalian cells has remained elusive. Here, we report that RTCB, the catalytic subunit of the tRNA ligase complex, and its co-factor archease mediate *XBP1* mRNA splicing both *in vitro* and *in vivo*. Depletion of RTCB in plasma cells of *Rtcb*<sup>fl/fl</sup> *Cd23*-Cre mice prevents XBP1s expression, which normally is strongly induced during plasma cell development. RTCB-depleted plasma cells show reduced and disorganized ER structures as well as severe defects in antibody secretion. Targeting RTCB and/or archease thus represents a promising strategy for the treatment of a growing number of diseases associated with elevated expression of XBP1s.

**Keywords** antibody secretion; archease; plasma cells; RTCB; *XBP1* mRNA splicing
**Subject Categories** Immunology; RNA Biology
The EMBO Journal (2014) 33: 2922–2936

See also: **W Filipowicz** (December 2014)

## Introduction

In mammalian cells, around 6% of all tRNAs are encoded as intron-containing pre-tRNA sequences that must undergo splicing in order to become active in protein translation (reviewed in Popow *et al*, 2012). tRNA splicing requires the tRNA ligase complex consisting of RTCB as the catalytic subunit, the DEAD-box helicase DDX1 and three subunits of unknown function: FAM98B, ASW and CGI-99 (Popow *et al*, 2011). Full enzymatic activity of RTCB depends on guanylation, which is provided by the co-factor archease working in cooperation with DDX1 (Popow *et al*, 2014).

In *Saccharomyces cerevisiae*, tRNA maturation is likewise catalyzed by the homologous tRNA ligase Trl1 altogether conducting three enzymatic reactions comprising hydrolysis of the 2′, 3′-cyclic phosphate to yield a 3′-hydroxyl (OH), 2′-phosphate terminus (cyclic phosphodiesterase activity), phosphorylation of the terminal 5′-OH group at the tRNA 3′ exon (GTP-dependent, RNA-polynucleotide kinase activity) and ligation of tRNA exon halves (ATP-dependent, RNA ligase activity) (Greer *et al*, 1983; Phizicky *et al*, 1986; Apostol *et al*, 1991; Sawaya *et al*, 2003). The phosphate incorporated into the newly formed phosphodiester bond therefore originates from the nucleotide triphosphate co-factor required for the kinase reaction (Greer *et al*, 1983; Phizicky *et al*, 1986). In contrast, the mammalian tRNA ligase RTCB directly joins 2′, 3′-cyclic phosphate and 5′-OH termini, leading to incorporation of the precursor-derived cyclic phosphate into the splice junction (Filipowicz & Shatkin, 1983; Filipowicz *et al*, 1983; Laski *et al*, 1983). This mechanism is referred to as 3′-5′ ligation. While the mammalian tRNA ligase complex is well characterized *in vitro*, less is known about its functions *in vivo*.

2′, 3′-cyclic phosphates and 5′-OH termini not only do characterize tRNA splicing but also are generated during unconventional splicing of *XBP1* mRNA as part of the unfolded protein response (UPR), a stress-signaling pathway activated upon accumulation of unfolded proteins in the ER lumen (reviewed in Hetz, 2012). Cytoplasmic splicing of *XBP1* mRNA is initiated by the ER transmembrane endonuclease IRE1 and is required for expression of the transcription factor XBP1s. Although in total there are three different UPR signaling branches in mammalian cells, the IRE1-XBP1 axis is the most ancient and conserved pathway and its improper functioning has been associated with many human diseases, such as cancer, autoimmunity and neurodegenerative disorders (reviewed in Hetz *et al*, 2013).

1 Institute of Molecular Biotechnology of the Austrian Academy of Sciences (IMBA), Vienna, Austria
2 European Molecular Biology Organization (EMBO), Heidelberg, Germany
3 Institute of Molecular Pathology (IMP), Vienna, Austria
4 European Molecular Biology Laboratory (EMBL), Heidelberg, Germany
5 Whitehead Institute for Biomedical Research, Cambridge, MA, USA
*Corresponding author. Tel: +43 1 79044 4840; E-mail: javier.martinez@imba.oeaw.ac.at
†These authors contributed equally to this work

Given its high conservation, studies in yeast critically contributed to the mechanistic understanding of unconventional mRNA splicing during the UPR. In *S. cerevisiae*, the endonuclease Ire1p directly senses perturbations in ER homeostasis and gains endoribonuclease activity to remove an intron from *HAC1* mRNA—the homologue of mammalian *XBP1* mRNA—that was retained after nuclear splicing. Cleavage by Ire1p generates mRNA exons displaying 2′, 3′-cyclic phosphate and 5′-OH termini, which are subsequently joined by the tRNA ligase Trl1 (Cox & Walter, 1996; Sidrauski *et al*, 1996; Sidrauski & Walter, 1997).

Similar to *HAC1* mRNA splicing in *S. cerevisiae*, the mRNA encoding for the mammalian transcription factor XBP1 retains a short, 26-nucleotide intron after canonical splicing, which is recognized and removed by activated IRE1 (Yoshida *et al*, 2001). Subsequent ligation of *XBP1* mRNA exon halves causes a frame shift that changes parts of the open reading frame and enables translation of XBP1s. In contrast to XBP1u, the protein product of unspliced *XBP1* mRNA, XBP1s is a potent transcription factor and regulates genes required to restore ER homeostasis such as chaperones or proteins involved in ER-associated protein degradation (ERAD) (Lee *et al*, 2003). Although unconventional splicing of *XBP1* mRNA resembles *HAC1* mRNA splicing in yeast, the mammalian RNA ligase involved in *XBP1* mRNA splicing has remained elusive.

A constitutively active UPR is a feature of specialized secretory cells (reviewed in Moore & Hollien, 2012). Antibody-secreting plasma cells for instance dramatically induce XBP1s expression during plasma cell differentiation from stimulated B cells (Reimold *et al*, 2001; Gass *et al*, 2002; Iwakoshi *et al*, 2003b), which coordinately induces changes in cellular structures to create a professional secretory phenotype enabling high rates of antibody production (Shaffer *et al*, 2004; McGehee *et al*, 2009; Taubenheim *et al*, 2012). Induction of the UPR in antibody-secreting cells differs from conventional UPR activation in that it is an integral part of the plasma cell differentiation program (Reimold *et al*, 2001; Iwakoshi *et al*, 2003b; Shaffer *et al*, 2004; Klein *et al*, 2006; Nera *et al*, 2006; Schmidlin *et al*, 2008). Accordingly, B cells deficient in XBP1 are unable to expand ER structures (Taubenheim *et al*, 2012), and mice lacking XBP1 show reduced serum immunoglobulin levels and impaired immunoglobulin response to immunization (Reimold *et al*, 2001; Iwakoshi *et al*, 2003b; Todd *et al*, 2009; Taubenheim *et al*, 2012). These findings suggest that induction of the UPR and XBP1 is required by plasma cells to achieve high rates of antibody secretion. Yet the role of XBP1 in plasma cells was proposed to extend beyond

its function in the UPR. Initial studies of mice with *Xbp1* deletion in the entire lymphoid system revealed that the absence of XBP1 does not only impact on antibody secretion but also severely affect plasma cell development (Reimold *et al*, 2001). However, more recent studies of a B-cell-specific conditional *Xbp1* mutant mouse model revealed either no or mild effects on plasma cell differentiation that were restricted to later stages of plasma cell development (Hu *et al*, 2009; Todd *et al*, 2009; Taubenheim *et al*, 2012). Additional roles for XBP1 have been proposed in the regulation of the plasma cell survival factor IL-6 (Iwakoshi *et al*, 2003b) and in homing of plasma cells to the bone marrow (Hu *et al*, 2009).

To analyze a possible function of the mammalian tRNA ligase complex in *XBP1* mRNA ligation, we depleted RTCB and its co-factor archease in HeLa cell lines and generated a mature B-cell-specific *Rtcb* knockout mouse. Data from these two models demonstrate an essential function of the tRNA ligase in *XBP1* mRNA splicing and the mammalian UPR and reveal a novel role of RTCB in supporting high rates of antibody secretion in plasma cells.

# Results

### An *in vitro* assay for *XBP1* mRNA splicing in HeLa cells

We established an *in vitro* splicing assay to monitor *XBP1* mRNA ligation using an internally radiolabeled human *XBP1* transcript encompassing the 26-nucleotide intron. This transcript is cleaved with recombinant, constitutively active IRE1 to form RNA fragments mimicking *XBP1* mRNA exon halves (Fig 1A and B). Upon addition of HeLa whole-cell extracts, these fragments were converted into a single, longer species representing the spliced form of *XBP1* mRNA (Fig 1A and B). Ligation activity was proportional to the protein concentration of cell extract added (Supplementary Fig S1A) and confirmed by splicing assays using either 5′ end- or 3′ end-labeled *XBP1* mRNA fragments (Supplementary Fig S1B and C).

Having established this assay, we depleted proteins with a potential role in *XBP1* mRNA splicing by RNAi and monitored the ligation activity in the resulting cell extracts. Since UPR-induced mRNA splicing is mediated by the tRNA ligase Trl1 in yeast (Sidrauski *et al*, 1996), we focused on the components of the human tRNA ligase complex (Popow *et al*, 2011, 2014). Indeed, depletion of its catalytic subunit RTCB largely impaired *in vitro* ligation of *XBP1* mRNA exon halves (Fig 1C). The same effect was seen after depletion of

**Figure 1.** ***In vitro* splicing of *XBP1* mRNA and subcellular localization of RTCB and archease.**

A   Schematic representation of the *in vitro* assay to monitor *XBP1* mRNA splicing. A radiolabelled human *XBP1* transcript encompassing the intron is pre-cleaved with recombinant, constitutively active IRE1 to form RNA fragments mimicking *XBP1* mRNA exon halves. Subsequent incubation with HeLa whole-cell extracts provides the ligation activity required to convert these fragments into a single, longer species representing the spliced form of *XBP1* mRNA.

B   An internally labeled fragment of *XBP1* mRNA including the intron (lane 1) was incubated with HeLa whole-cell extracts (Wce, lanes 4–7) or pre-cleaved with recombinant IRE1 endonuclease and afterward supplemented with buffer (lanes 8–11) or Wce (lanes 12–15) for the indicated time periods. After addition of Wce, cleaved *XBP1* mRNA fragments were efficiently converted into the spliced form *XBP1* mRNA (compare to lane 2). A nucleotide (nt) size marker is shown in lane 3. An unspecific band is marked with an asterisk.

C   HeLa cells were transfected with control siRNA (siGFP) or siRNAs against *RTCB*, archease or both and harvested 3 days post-transfection. Whole-cell extracts were incubated with a 3′ end-labeled *XBP1* mRNA pre-cleaved by recombinant IRE1 for 15 min.

D   Subcellular localization of RTCB and archease assessed by Western blot analysis of fractions obtained after subcellular fractionation of HeLa cells treated with 300 nM thapsigargin (Tg) for the indicated time periods. HSP90 (cytoplasm), calnexin (membranes) and lamin A/C (nucleus) were used as marker proteins for the individual fractions collected (*n* = 5).

E   Subcellular localization of RTCB and archease visualized by immunofluorescence staining of HeLa cells treated with 300 nM Tg for the indicated time periods. The nucleus is visualized by DAPI staining. Calnexin staining is used to mark the ER membrane (*n* = 4).

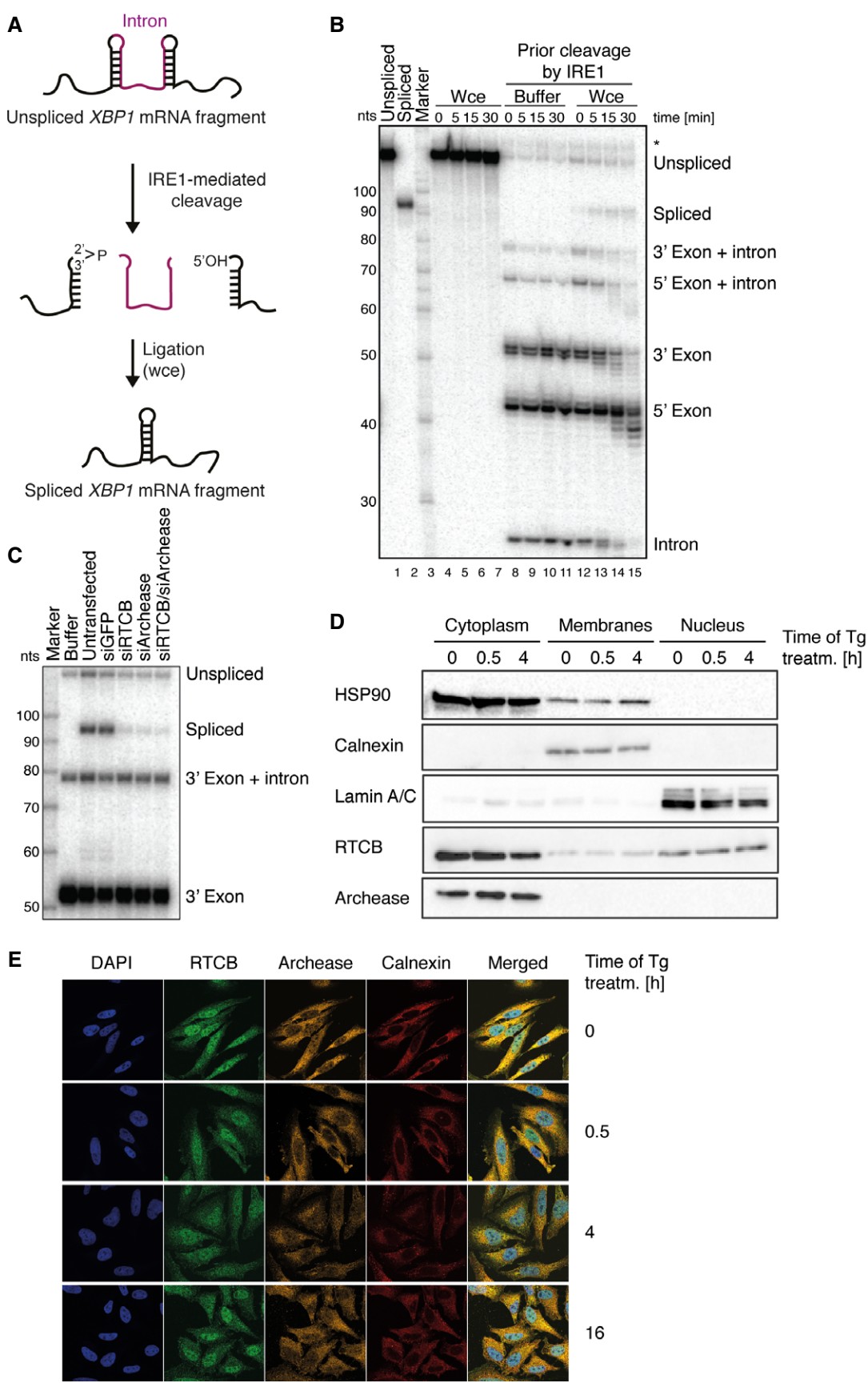

**Figure 1.**

archease or both proteins (Fig 1C), while addition of recombinant, wild-type archease but not of catalytically inactive archease mutants stimulated the RNA ligation activity in wild-type cell extracts (Popow *et al*, 2014) (Supplementary Fig S1D). Taken together, these results argue that both RTCB and archease are required for ligation of *XBP1* mRNA exon halves *in vitro*.

## RTCB and archease localize to the cytoplasm of HeLa cells

Mammalian tRNA splicing is thought to be a predominantly nuclear process (De Robertis *et al*, 1981; Nishikura & De Robertis, 1981; Lund & Dahlberg, 1998), while unconventional splicing of *XBP1* mRNA takes place in the cytoplasm (Cox *et al*, 1993; Sidrauski *et al*, 1996; Sidrauski & Walter, 1997; Yanagitani *et al*, 2009). To address whether the subcellular localization of RTCB and archease is compatible with a role in both processes, we performed subcellular fractionation experiments (Fig 1D) and immunofluorescence staining (Fig 1E) in control cells and upon UPR induction by means of thapsigargin (Tg) treatment, an inhibitor of ER $Ca^{2+}$-ATPases. We detected RTCB both in the nucleus and in the cytoplasm, which is in agreement with a recent report identifying the tRNA ligase as part of RNA transport complexes shuttling between these two compartments (Perez-Gonzalez *et al*, 2014). In contrast, archease was strongly enriched in the cytoplasm (Fig 1D) and found in perinuclear regions stained by the ER membrane marker calnexin (Fig 1E). This subcellular distribution of both proteins was stable and did not change after Tg treatment (Fig 1D and E). Thus, while no active recruitment to the site of *XBP1* mRNA splicing occurs upon UPR induction, a substantial fraction of RTCB and archease constitutively localizes to the vicinity of the ER membrane and could therefore function in cytoplasmic *XBP1* mRNA ligation in living cells.

## Simultaneous depletion of RTCB and archease abolishes *XBP1* mRNA splicing in cell culture

To test whether RTCB or archease facilitates ligation of endogenous *XBP1* mRNA in HeLa cells, we efficiently depleted both proteins by doxycycline (Dox)-inducible expression of small hairpin RNAs (shRNAs) using the miR-E backbone (Fellmann *et al*, 2013) (Fig 2A). As reported before (Popow *et al*, 2011), depletion of RTCB also led to a simultaneous depletion of DDX1 and FAM98B (Supplementary Fig S2A), two subunits of the tRNA ligase complex. Following UPR induction, reduced XBP1s expression was mainly detected in archease-depleted cells (Fig 2A). In contrast to RTCB, archease does not possess any RNA ligation activity and is not constitutively

associated with the tRNA ligase complex (Popow *et al*, 2014). Consequently, affinity purifications of FLAG-archease showed no detectable RNA ligation activity in *XBP1* mRNA splicing assays unlike immunoprecipitations of FLAG-RTCB and FLAG-DDX1 (Supplementary Fig S1E). RTCB-depleted HeLa cells, however, supported XBP1s expression almost to wild-type levels (Fig 2A). As revealed by RT–PCR experiments using primers flanking the non-conventional splice sites, also the amount of *XBP1s* mRNA was reduced upon archease knockdown, resulting in a decreased ratio of spliced (*XBP1s*) to unspliced (*XBP1u*) mRNA in comparison with control samples (Fig 2B and Supplementary Fig S2B). While in apparent contrast to our *in vitro* splicing assays, these results are in agreement with earlier studies showing that RNAi-mediated depletion of RTCB alone does not impair *XBP1* mRNA splicing (Iwawaki & Tokuda, 2011). Therefore, archease appears to be crucial for
non-conventional splicing of *XBP1* mRNA as its stimulatory activity (Popow *et al*, 2014) sustains ligation activity in the presence of reduced amounts of RTCB.

Since depletion of neither RTCB nor archease sufficed to fully abrogate XBP1s induction, we simultaneously depleted both proteins in HeLa cells. After Tg treatment, XBP1s was no longer detectable at the protein level (Fig 2C) and greatly reduced at the mRNA level (Fig 2D and Supplementary Fig S2C). This result was supported by RT–qPCR experiments showing a clear reduction in *XBP1s* mRNA expression (Fig 2E). As the transcription factor XBP1s auto-regulates its own promoter (Yoshida *et al*, 2001; Lee *et al*, 2002), the levels of total *XBP1* mRNA (Fig 2F) and *XBP1u* mRNA (Fig 2G) likewise failed to accumulate, especially at later stages of UPR signaling. Thus, sufficient inhibition of tRNA ligase activity can only be achieved by simultaneous targeting of archease.

Accumulation of XBP1s leads to the transcriptional activation of downstream target genes such as the ERAD component *EDEM1* and the co-chaperone *DNAJB9* (Lee *et al*, 2003), which serve to decrease the load of unfolded proteins within the ER. We observed increased expression of *EDEM1* and *DNAJB9* mRNA in control cells after 8–16 h of Tg treatment (Fig 2H and I). This response was abolished in cells simultaneously depleted of both, RTCB and archease. In contrast, we detected a less strong reduction in the expression of the general stress responders HSPA5 (BiP), a member of the HSP70 family, and the pro-apoptotic transcription factor CHOP (Supplementary Fig S2D and E), both of which depend on the activation of other branches of UPR signaling and thus are less susceptible to changes in XBP1s levels (Yoshida *et al*, 2001; Lee *et al*, 2003; Chen *et al*, 2014). More importantly, the moderately affected expression

**Figure 2.  Simultaneous depletion of RTCB and archease abolishes XBP1s expression in cell culture.**

A, B  Tetracycline-inducible (Tet-ON) HeLa cells were incubated with 1 µg/ml doxycycline (Dox) for six consecutive days to stimulate expression of shRNAs targeting *RTCB*, archease or non-targeting control followed by treatment with 300 nM Tg for the indicated time periods. Induction of XBP1s (XBP1 spliced) expression was monitored by Western blot (A) and RT–PCR (B) analysis. The relative contribution of *XBP1s* mRNA to total levels of *XBP1* mRNA was analyzed by densitometry (*n* = 3).

C, D  Tet-ON HeLa cells expressing shRNAs targeting *RTCB* and archease or a control cell line expressing two copies of the control shRNA were treated and analyzed as in (A, B) (*n* = 5).

E–I  Tet-ON HeLa cells expressing shRNAs targeting *RTCB* and archease or a control cell line expressing two copies of the control shRNA were treated with Dox (1 µg/ml, 6 days) and Tg (300 nM, 24-h time course). Relative mRNA levels of *XBP1s* and *XBP1u* as well as total *XBP1* mRNA and induction of *EDEM1* and *DNAJB9* mRNA were analyzed by RT–qPCR (*n* = 5, mean expression levels and SEM are displayed). Expression levels were normalized to *ACTB* mRNA levels and to the untreated control sample. Two-way ANOVA was used to analyze the statistical significance of differences in mRNA levels between control and RTCB/archease-depleted cells (*$P < 0.05$, **$P < 0.01$, ***$P < 0.001$, ****$P < 0.0001$).

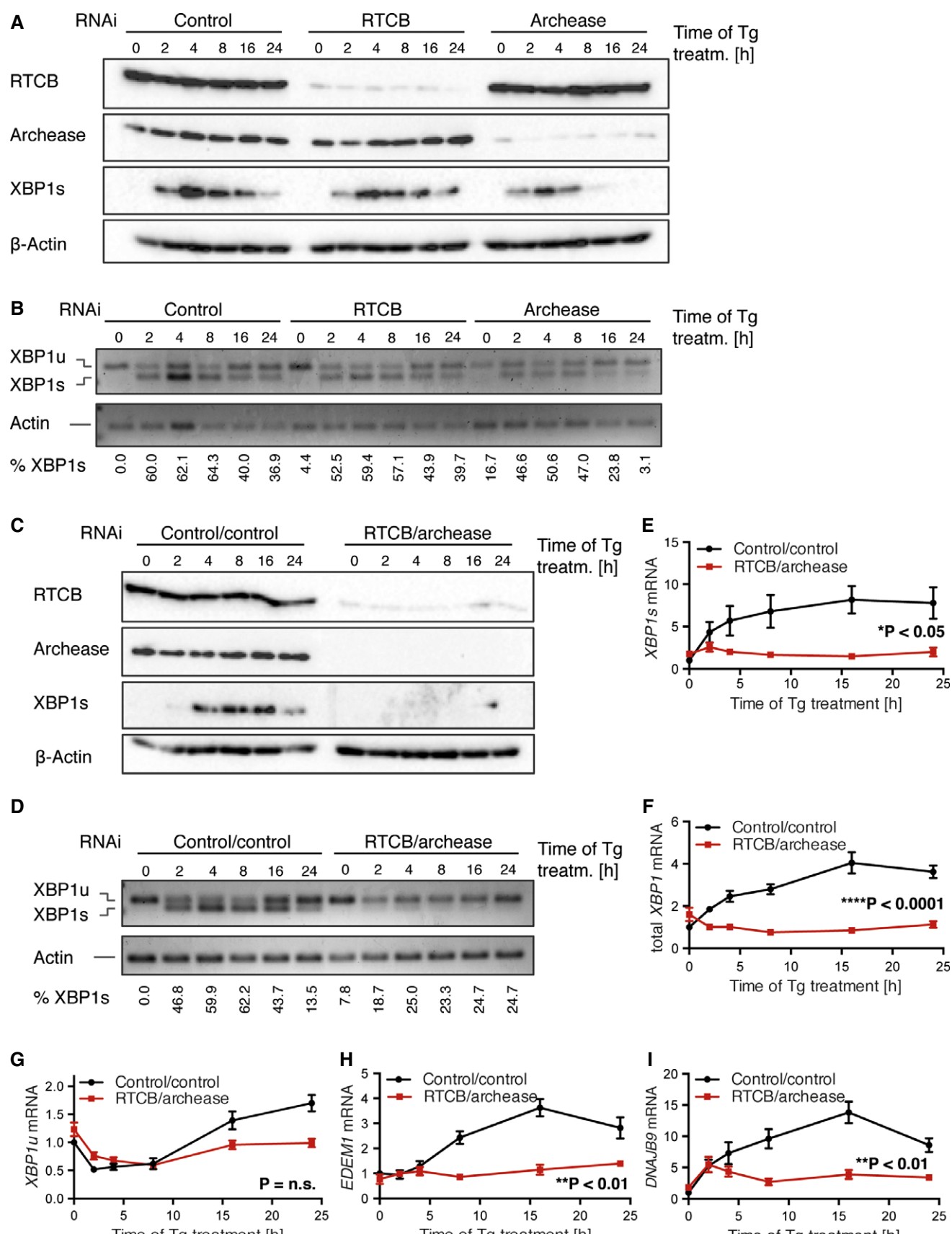

**Figure 2.**

of CHOP upon depletion of RTCB and archease suggests that inhibition of tRNA ligase activity does not impair the induction of apoptosis under conditions of prolonged ER stress (Hetz, 2012). Likewise, the levels of *BLOS1* and *PDGFRB* mRNAs, known substrates of regulated IRE1-dependent decay (RIDD), an mRNA degradation pathway initiated by IRE1-mediated cleavage (Hollien & Weissman, 2006; Hollien *et al*, 2009), remained unchanged after RTCB and archease depletion and were equally reduced after induction of the UPR (Supplementary Fig S2F and G). This result confirms that depletion of RTCB and archease in the context of UPR activation does not interfere with the endonucleolytic activity of IRE1, but specifically disrupts *XBP1* mRNA splicing and thus the induction of XBP1s-specific downstream target genes.

RTCB and archease have been linked to tRNA splicing and thus to the maturation of intron-containing pre-tRNAs in eukaryotes and in archaebacteria (Popow *et al*, 2011, 2014; Desai *et al*, 2014). Although only a subset of all tRNAs are encoded by intron-containing pre-tRNA sequences, each organism possesses at least one tRNA isoacceptor family of which all or almost all members depend on splicing in order to become functional in translation. In humans, these include Ile-TAT, Arg-TCT, Tyr-ATA and Tyr-GTA. Using probes specifically recognizing only splicing-dependent tRNAs (Ile-tRNAs and Arg-tRNAs), we observed a decrease in the levels of mature transcripts as a consequence of RTCB and archease depletion (Supplementary Fig S3A and B). In contrast, levels of splicing-independent methionine tRNAs remained unchanged (Supplementary Fig S3A and B) as were global protein translation rates measured by metabolic labeling ($^{35}$S-methionine and $^{35}$S-cysteine) (Supplementary Fig S3C and D). These results thus indicate that the reduced levels of splicing-dependent mature tRNAs do not lead to a global defect in protein synthesis in RTCB- and archease-depleted cells.

## RTCB is required for the generation of XBP1s during plasma cell differentiation

Apart from being caused by stress agents that perturb ER functions, the UPR can also arise as a part of a developmental program that is initiated during the differentiation of secretory cells. This includes differentiation of activated B cells into antibody-secreting plasma cells. Given the newly identified function of RTCB in *XBP1* mRNA splicing in cell culture, we next studied the *in vivo* function of RTCB during plasma cell differentiation. To this end, we established a B-cell-specific mouse knockout model—*Rtcb*^fl/fl^ *Cd23*-Cre—which initiates Cre-mediated deletion in immature B cells of the spleen and leads to efficient gene deletion in all mature B-cell types (Kwon *et al*, 2008). We determined the efficiency of *Rtcb* deletion in B220-enriched splenocytes by PCR genotyping, which demonstrated an almost complete deletion of *Rtcb* in *Rtcb*^fl/fl^ *Cd23*-Cre B cells (Supplementary Fig S4A). As a result, the RTCB protein was almost absent in *Rtcb*^fl/fl^ *Cd23*-Cre B cells, together with diminished levels of other tRNA ligase complex members, such as DDX1 and FAM98B, but not CGI-99 (Supplementary Fig S4B). Flow cytometric analysis showed that the different B-cell subsets were present in similar numbers in the spleen of *Rtcb*^fl/fl^ *Cd23*-Cre, *Rtcb*^fl/+^ *Cd23*-Cre and control mice (Supplementary Fig S4C and D).

We next investigated the role of RTCB in the splicing of *Xbp1* mRNA and the expression of XBP1s protein during plasma cell differentiation. For this purpose, we isolated splenic B cells from control (*Rtcb*^fl/fl^ or *Rtcb*^fl/+^), heterozygous (*Rtcb*^fl/+^ *Cd23*-Cre) or homozygous (*Rtcb*^fl/fl^ *Cd23*-Cre) mice and stimulated them with lipopolysaccharide (LPS) for 4 days, which induced differentiation via activated B cells and pre-plasmablasts to plasmablasts. By immunoblot analysis, we observed a transient increase in RTCB and DDX1 protein expression in wild-type cells at day 2 of LPS stimulation (compared to β-actin expression), whereas RTCB protein was absent in *Rtcb*^fl/fl^ *Cd23*-Cre cells during the entire culture period (Fig 3A). Again, we noticed a concomitant depletion of other tRNA ligase complex members, such as DDX1, FAM98B or CGI-99 (Fig 3A). Furthermore, we analyzed the induction of XBP1 protein expression during LPS stimulation (d0–d4) (Fig 3B). In wild-type and heterozygous B cells, both the inactive XBP1u and the active XBP1s protein were readily induced by day 2, whereas RTCB-deficient cells up-regulated XBP1u only transiently and failed to produce XBP1s (Fig 3B). At day 4, cells present in the LPS cultures can be subdivided according to the surface expression of CD22 and CD138 into activated B cells (CD138^−^ CD22^+^), pre-plasmablasts (CD138^− CD22^low^) and plasmablasts (CD138^+^ CD22^−^), which secrete high amounts of antibodies (M. Minnich and M. Busslinger, unpublished observation). We evaluated XBP1s levels in sorted RTCB-deficient pre-plasmablasts and plasmablasts (Fig 3C). These cell populations showed significantly decreased levels of XBP1s. Interestingly, we observed incomplete depletion of RTCB in the differentiated *Rtcb*^fl/fl^ *Cd23*-Cre plasmablasts (Fig 3C), indicating that the absence of RTCB in plasmablasts is not well tolerated, thus selecting for non-deleted cells. In addition to XBP1s protein, we also studied *Xbp1* mRNA levels and found that, similar to *Rtcb* mRNA, both *Xbp1s* and total *Xbp1* mRNAs were significantly reduced in unfractionated RTCB-deficient cells at day 4 of LPS stimulation (Fig 3D). We also observed decreased mRNA levels of the XBP1 target gene *Edem1* (Yoshida *et al*, 2003). It was previously reported that expression of the secreted (μS) but not the membrane bound form (μM) of the Igμ heavy chain depends on XBP1s (Taubenheim *et al*, 2012; Benhamron *et al*, 2013). We therefore investigated the presence of these transcripts and found that IgμS transcripts were strongly decreased, whereas IgμM mRNA levels were only minimally affected (Fig 3D).

## RTCB is required for immunoglobulin secretion by plasmablasts *in vitro*

After *in vitro* stimulation with LPS, *Rtcb*^fl/fl^ *Cd23*-Cre B cells were able to develop into CD138^+^ CD22^−^ plasmablasts and CD138^− CD22^low^ pre-plasmablasts, although the percentages of CD138^+^ CD22^−^ plasmablasts were slightly reduced as compared to wild-type or heterozygous controls (Supplementary Fig S5A). This reduction was likely caused by a defect in proliferation rather than differentiation (Supplementary Fig S5B). As measured by ELISA, IgM levels were significantly reduced in culture supernatants of RTCB-deficient pre-plasmablasts and plasmablasts (Fig 4A). ELISPOT analysis of sorted plasmablasts revealed similar numbers of IgM-secreting cells for both the control and experimental genotypes (Fig 4B). However, in this assay, *Rtcb*^fl/fl^ *Cd23*-Cre plasmablasts gave rise to significantly smaller sports than control plasmablasts, indicating that these cells secrete lower levels of immunoglobulins in the absence of RTCB (Fig 4B). In summary, these data indicate that the loss of RTCB strongly interferes with the capacity of plasmablasts to secrete

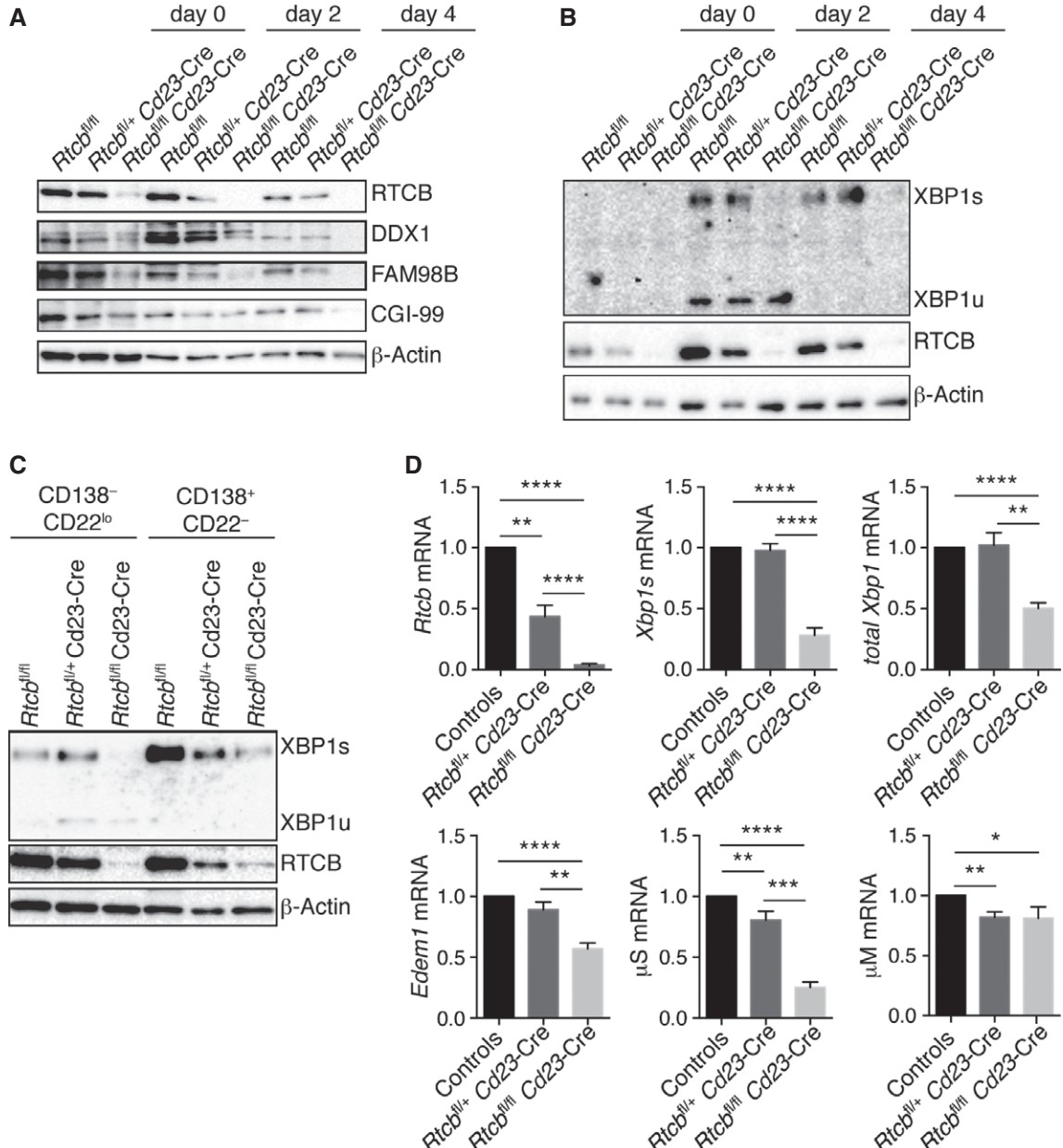

**Figure 3. RTCB is required for induction of XBP1s during plasma cell differentiation.**

B220[+] splenocytes of control (*Rtcb*[fl/fl] or *Rtcb*[fl/+]), *Rtcb*[fl/+] *Cd23*-Cre or *Rtcb*[fl/fl] *Cd23*-Cre mice were stimulated with 20 μg/ml LPS for 4 days.

A, B   Protein levels of RTCB, tRNA ligase complex members (DDX1, FAM98B, CGI-99) and XBP1 were monitored by Western blot analysis (*n* > 3).

C   FACS-sorted CD138[+] CD22[−] plasmablasts or CD138[−] CD22[low] pre-plasmablasts were probed for expression of the indicated proteins by Western blot analysis (*n* = 2).

D   Relative mRNA levels of *Rtcb*, *Xbp1s*, total *Xbp1*, *Edem1*, μM and μS were analyzed by RT–qPCR in fractionated LPS-stimulated cells at day 4 (*n* = 4, mean expression levels and SEM are displayed). Expression levels were normalized to *Actb* mRNA levels and to B cells from control mice. An unpaired Student's *t*-test was used to analyze the statistical significance of differences in mRNA levels (*$P < 0.05$, **$P < 0.01$, ***$P < 0.001$, ****$P < 0.0001$).

antibodies, although the ability of RTCB-deficient B cells to differentiate into plasmablasts remains largely intact.

Activation of XBP1 during differentiation of plasma cells was shown to increase the capacity of the ER, preparing these cells to cope with the enormous amount of immunoglobulin production and secretion (Taubenheim *et al*, 2012). We analyzed the ER morphology of RTCB-deficient plasmablasts by electron microscopy and found disorganized and less dense ER structures as compared to wild-type plasmablasts, which displayed an extensively layered and organized ER, characteristic of antibody-secreting plasma cells (Fig 4C). To exclude that the reduced antibody-secreting capacity of RTCB-deficient plasmablasts is caused by defects in global protein synthesis, we analyzed mature tRNA levels and protein synthesis rates of *Rtcb*[fl/fl] *Cd23*-Cre cells after 4 days of LPS stimulation. In

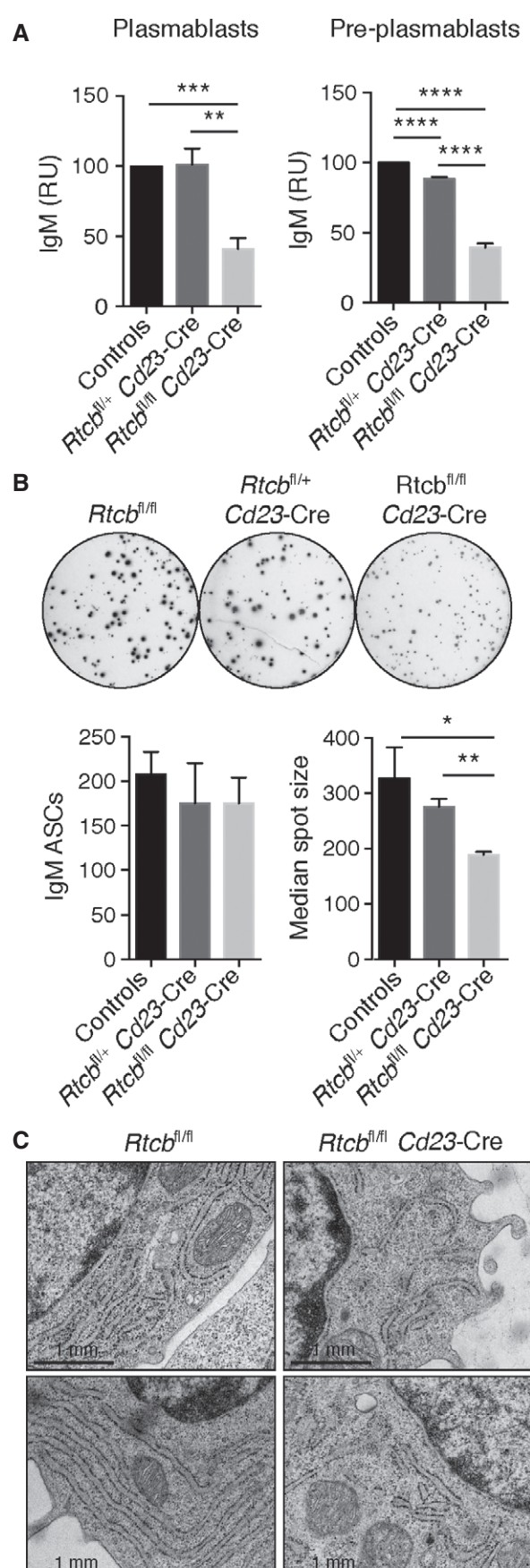

**Figure 4.** RTCB is required for immunoglobulin secretion by antibody-secreting cells *in vitro*.

B220$^+$ cells were enriched from the spleen of control (*Rtcb*$^{fl/fl}$ or *Rtcb*$^{fl/+}$), *Rtcb*$^{fl/+}$ *Cd23*-Cre or *Rtcb*$^{fl/fl}$ *Cd23*-Cre mice and cultured in the presence of 20 μg/ml LPS for 3 days.

A   IgM ELISA. Identical numbers of FACS-sorted CD138$^+$ CD22$^-$ plasmablasts or CD138$^-$ CD22$^{low}$ pre-plasmablasts of the indicated genotypes were plated for 24 h prior to ELISA analysis of their supernatants. The data are presented as relative units (RU) compared to control cells (*n* = 4, mean and SEM are displayed). An unpaired Student's *t*-test was used to analyze the statistical significance (\*\**P* < 0.01, \*\*\**P* < 0.001, \*\*\*\**P* < 0.0001).

B   IgM ELISPOT analysis. FACS-sorted CD138$^+$ CD22$^-$ plasmablasts (500 cells) were plated for 16–18 h. A representative assay is shown in the top panel. Bar diagrams in the low panel show the average number of IgM-secreting cells and their median spot size (measured in pixels), respectively (*n* = 4 mean and SEM are displayed). An unpaired Student's *t*-test was used to analyze the statistical significance (\**P* < 0.05, \*\**P* < 0.01).

C   Plasmablasts were analyzed by electron microscopy. Representative plasmablasts of the indicated genotypes are shown.

mouse, all genes encoding for the isoacceptor families Tyr-GTA and Leu-CAA contain introns and thus have to be spliced in order to give rise to mature tRNAs. Analyzing these two splicing-dependent tRNA families, we found that mature Tyr- and Leu-tRNAs, but not splicing-independent Met-tRNAs, were reduced but still present in RTCB-depleted plasmablasts (Supplementary Fig S6A). However, despite a decreased level of mature tRNAs, *Rtcb*$^{fl/fl}$ *Cd23*-Cre cells did not display global changes in protein synthesis rates as shown by unchanged levels of $^{35}$S-methionine and $^{35}$S-cysteine incorporation after metabolic labeling (Supplementary Fig S6B). In LPS-stimulated *Rtcb*$^{fl/fl}$ *Cd23*-Cre cells, we could, however, reproducibly observe a decreased intensity of a single band of ~75 kDa that corresponds to the expected size of the secreted Igμ heavy chain.

## RTCB-deficient B cells differentiate into plasma cells with an impaired capacity to secrete immunoglobulins *in vivo*

We finally characterized the role of RTCB in the generation and function of antibody-secreting cells *in vivo*. To this end, we examined plasma cell differentiation during a T-cell-independent immune response by immunizing mice with trinitrophenyl (TNP)-coupled LPS. At day 14 after immunization, we observed equal numbers of plasma cells (CD28$^+$ CD138$^+$ Lin$^-$) in the spleen of immunized control *Rtcb*$^{fl/fl}$ and *Rtcb*$^{fl/fl}$ *Cd23*-Cre mice (Fig 5A). However, the number of TNP-specific antibody-secreting cells was reduced in *Rtcb*$^{fl/fl}$ *Cd23*-Cre mice when assessed by ELISPOT assay (Fig 5B). The size of the spots was also reproducibly smaller, indicating low immunoglobulin production by RTCB-deficient antibody-secreting cells (Fig 5B). Likewise, TNP-specific serum immunoglobulin levels measured by ELISA were significantly reduced in *Rtcb*$^{fl/fl}$ *Cd23*-Cre mice (Fig 5C). Together, these data strengthen our *in vitro* results and confirm that the ability of RTCB-deficient B cells to generate plasma cells remains largely intact while their capacity to secrete immunoglobulins is strongly affected.

## Discussion

In this study, we have shown that RTCB, the catalytic subunit of the mammalian tRNA ligase complex, mediates ligation of *XBP1* mRNA

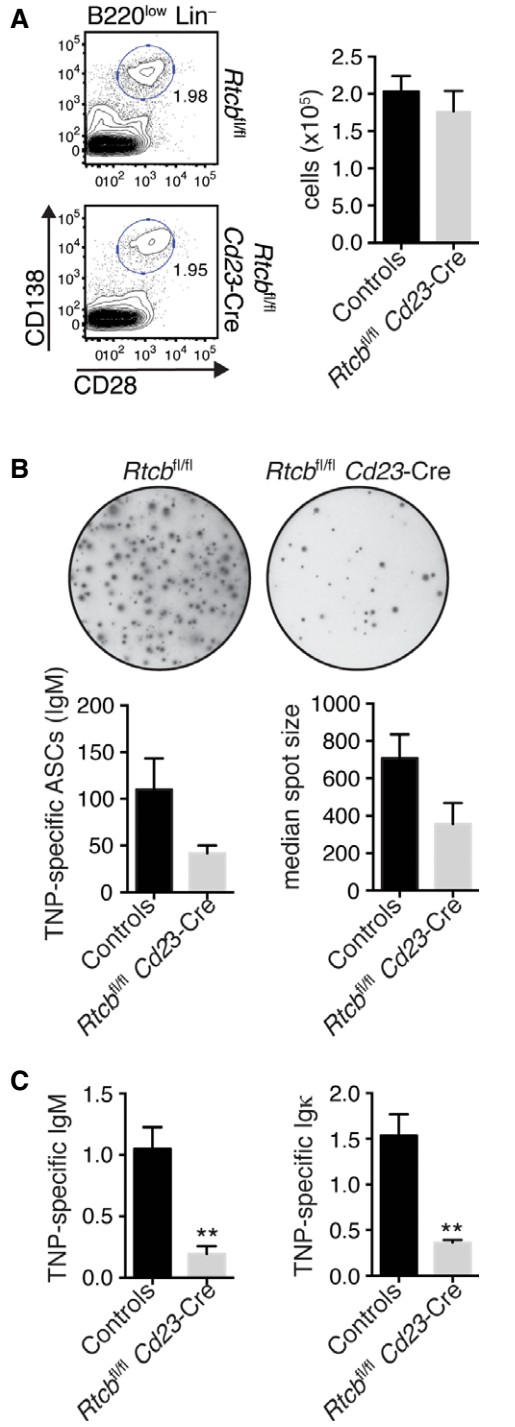

**Figure 5. RTCB-deficient B cells show an impaired capacity to secrete immunoglobulins *in vivo*.**

Control (*Rtcb*^fl/fl or *Rtcb*^fl/+) and *Rtcb*^fl/fl *Cd23*-Cre mice were injected intraperitoneally with 50 μg of TNP-(0.5)-LPS and analyzed 2 weeks after immunization.

A  Plasma cell numbers in the spleen were determined by flow cytometry. Representative contour plots are shown. Bar diagrams represent total plasma cell numbers (*n* = 4, mean and SEM are displayed). Plasma cells were defined as CD28⁺ CD138⁺ B220^low Lin⁻ (CD4⁻ CD8⁻ CD21⁻ F4/80⁻).

B  IgM ELISPOT analysis of MACS-enriched CD138⁺ cells after plating identical numbers for 16–18 h. A representative assay is shown in the upper panel. Bar diagrams show the average number of IgM-secreting cells and their median spot size (measured in pixels), respectively (*n* = 3 mean and SEM are displayed).

C  The serum titers of TNP-specific IgM and Igκ were determined by ELISA (*n* = 4 mean and SEM are displayed). An unpaired Student's *t*-test was used to analyze the statistical significance of differences (**P < 0.01).

well as in genetically engineered mouse ES cells (Lu *et al*, 2014), only a minor effect was seen after shRNA-mediated RTCB depletion in HeLa cells. This result is in line with a previous report showing no decrease in *XBP1* mRNA splicing efficiency upon knockdown of RTCB by means of siRNAs (Iwawaki & Tokuda, 2011). We therefore propose that a few ligase complexes—probably associated with the ER membrane—may suffice to splice *XBP1* mRNA upon induction of ER stress as long as archease is present to stimulate enzymatic rates (Popow *et al*, 2014). Hence, full impairment of XBP1s induction can only be achieved by RNAi-mediated depletion of both, RTCB and its co-factor archease, as indicated by our data.

The important function of archease during the UPR is further supported by its subcellular distribution. Even though tRNA splicing is thought to be a predominantly nuclear process (De Robertis *et al*, 1981; Nishikura & De Robertis, 1981; Lund & Dahlberg, 1998), we found archease and the majority of RTCB localizing to cytoplasmic compartments. This subcellular distribution of RTCB coincides with a recent report describing the tRNA ligase as part of an RNA transport complex shuttling between nucleus and cytoplasm (Perez-Gonzalez *et al*, 2014). This flexible localization of RTCB and the cytoplasmic distribution of archease suggest that shuttling of the tRNA ligase not only supports RNA transport but also enables guanylation of RTCB by archease. Furthermore, localization of RTCB in the cytoplasm allows interaction with IRE1α at the ER membrane (Lu *et al*, 2014). Given its importance for full enzymatic activity of the tRNA ligase complex, we speculate that also archease might—directly or indirectly—associate with IRE1 and therefore localize to foci of increased *XBP1* mRNA splicing.

We chose the antibody-secreting capacity of plasma cells, normally characterized by chronic activation of the UPR and high levels of XBP1s expression, as a physiological example to confirm the role of RTCB in *XBP1* mRNA splicing *in vivo*. To this end, we generated a conditional RTCB knockout mouse model, *Rtcb*^fl/fl *Cd23*-Cre, in which RTCB is specifically deleted in mature B cells. Neither overall B-cell numbers nor plasma cell differentiation were affected in these mice *in vivo*. However, we detected slightly decreased percentages of plasmablasts after *ex vivo* LPS stimulation of RTCB-depleted B cells. According to previous reports, proliferation and plasma cell differentiation are tightly linked, in that the probability of activated B cells to develop into plasma cells increases with the number of cell divisions (Hasbold *et al*, 2004). Using cell trace experiments, we observed that most wild-type cells that

exon halves during the UPR. We extended this finding and revealed the role of RTCB in the differentiation of B cells into plasma cells *in vivo*, a process that is characterized by *XBP1* mRNA splicing and physiological induction of the UPR.

Moreover, we described the critical function of archease, a co-factor of the tRNA ligase complex (Popow *et al*, 2014), in *XBP1* mRNA splicing. Recent work by Lu *et al* has reported similar findings. While full depletion of RTCB alone was enough to block XBP1s expression in genetically engineered plasma cells (this study) as

became plasmablasts had undergone multiple rounds of cell divisions, while *Rtcb*[fl/fl] *Cd23*-Cre plasmablasts had completed a significantly lower number of division cycles, Thus, indirect effects caused by the decreased proliferation rates observed in RTCB-deficient cells, rather than specific defects in plasma cell lineage choices, might contribute to the decrease in plasmablasts detected in RTCB-depleted B-cell cultures *in vitro*. As deficiency in XBP1 itself does not affect B-cell proliferation (Todd *et al*, 2009; Taubenheim *et al*, 2012), the reduced proliferation rates seen in *Rtcb*[fl/fl] *Cd23*-Cre B cells are most probably due to functions of RTCB that are unrelated to *Xbp1* mRNA splicing. In line with this, a recent report described reduced proliferation rates in RTCB-depleted unstressed ES cells, a condition in which XBP1s is not expressed (Lu *et al*, 2014).

Besides dramatic changes in ER morphology, RTCB-depleted plasma cells showed reduced rates of antibody secretion both *in vitro* and *in vivo* and thus resembled the phenotype previously observed in a B-cell-specific *Xbp1* knockout mouse (Hu *et al*, 2009; Todd *et al*, 2009; Taubenheim *et al*, 2012). Although our data strongly suggest that the defect in antibody secretion seen in RTCB-deficient plasma cells is a result of their inability to generate XBP1s, we cannot exclude that additional defects contribute to this phenotype. Due to its implication in tRNA splicing, deficiencies in global protein synthesis might be expected, leading to decreased production and secretion of proteins, including IgM. However, in contrast to a recent report (Lu *et al*, 2014), we did not observe any defect in global protein synthesis in *ex vivo* stimulated RTCB-deficient plasmablasts. This discrepancy might arise from the different cell systems used, that is, pluripotent ES cells versus differentiated B cells. Since tRNAs have been reported to be extremely stable with half-lives of weeks, it is tempting to speculate that fully differentiated or non-proliferating cells are able to maintain mature tRNA levels constant over longer periods of time and that cell division numbers rather than time after RTCB depletion might crucially influence abundance of mature tRNAs and consequently protein synthesis rates.

Collectively, our data show that RTCB together with its cofactor archease mediates *XBP1* mRNA splicing during the UPR. Upon RTCB depletion, plasma cells fail to induce expression of XBP1s during differentiation. They are also unable to expand ER structures and show reduced rates of antibody secretion both *in vitro* and *in vivo*. These findings constitute the first report of an *in vivo* function of the mammalian tRNA ligase complex. Furthermore, the identification of the long-sought *XBP1* mRNA splicing ligase opens new avenues in the treatment of a growing number of diseases associated with elevated levels of XBP1s expression such as multiple myeloma (Nakamura *et al*, 2006; Carrasco *et al*, 2007; Chapman *et al*, 2011), triple-negative breast cancer (Chen *et al*, 2014), pre-B-cell acute lymphoblastic leukemia (ALL) (Kharabi Masouleh *et al*, 2014) and B-cell chronic lymphocytic leukemia (CLL) (Tang *et al*, 2014).

# Materials and Methods

### Cell culture and siRNA transfection

HeLa cells were cultured at 37°C, with 5% $CO_2$ in 1× Dulbecco's modified Eagle's medium (Invitrogen) supplemented with 10% fetal bovine serum (Sigma), 3 mM glutamine (Sigma), 100 U/ml

penicillin and 100 μg/ml streptomycin sulfate (Sigma). For lenti- or retroviral packaging, LentiX (Clontech) or PlatE cells (CellBiolabs) were cultured as described above. siRNA transfections were performed using Lipofectamine 2000 reagent (Invitrogen) according to the manufacturer's instructions. For *in vitro* studies, archease and RTCB were depleted using ON-TARGETplus siRNAs (Dharmacon).

### Preparation of whole-cell extracts

HeLa cells were grown to confluency and harvested in 1× lysis buffer (30 mM HEPES pH 7.4, 100 mM KCl, 5 mM $MgCl_2$, 10% (v/v) glycerol, 0.2% (v/v) Nonidet P-40, 0.5 mM DTT, 0.1 mM phenylmethylsulfonyl fluoride (PMSF)) supplemented with Phosphatase Inhibitor Cocktail Set II (Merck). Extracts were diluted to 3 μg/μl total protein concentration unless otherwise indicated.

### Preparation of *XBP1* mRNA fragment

Human *XBP1* pre-mRNA fragment was transcribed using AmpliScribe T7-Flash (Epicentre) according to the manufacturer's protocol. Templates for the spliced and unspliced form of human *XBP1* mRNA were obtained by PCR on genomic DNA and cDNA, respectively, using the following primers: T7-hXBP118FW, 5′-TAA TAC GAC TCA CTA TAG GGG AAT GAA GTG AGG CCA GT-3′ and hXBP118RV, 5′-AAT CCA TGG GGA GAT GTT CTG GAG-3′. Sequence was confirmed by sequencing. The recovered transcripts were dissolved in 1× RNAi buffer (30 mM HEPES pH 7.4, 100 mM KCl, 5 μM $MgCl_2$, 0.5 μM DTT, 10 (v/v) % glycerol) and annealed to end-matching LNA oligos (1 μM) (to minimize exonucleolytic cleavage upon incubation with cell extracts) by 30 s incubation at 95°C followed by cooling to room temperature. The LNA-modified oligos hXBP-5 (5′-CAT TCC C-3′) and hXBP-3 (5′-AAT CCA G-3′) were obtained from Exiqon. Internally labeled *XBP1* mRNA transcript was synthesized in the presence of $^{32}$P-αGTP (Perkin Elmer). For 5′ end labeling, RNA was dephosphorylated using calf intestinal phosphatase (NEB), treated with proteinase K and subsequently labeled using T4 polynucleotide kinase (NEB) and $^{32}$P-γATP (Perkin Elmer). All reactions were performed as described by the manufacturer. RNA 3′ end labeling was achieved by direct ligation to $^{32}$P-pCp (Perkin Elmer) using RNA ligase I (NEB), according to manufacturer's protocol. All labeled transcripts were purified by preparative PAGE and dissolved in 1× RNAi buffer.

### *In vitro* RNA splicing assay

LNA-stabilized *XBP1* pre-mRNA fragment (0.1 μM) was preincubated with recombinant IRE1 (Volkmann *et al*, 2011) for 5 min in 1× tRNA ligation buffer (400 mM KCl, 125 mM spermidine, 20 mM ATP, 20 mM GTP, 10 mM DTT, 25 mM $MgCl_2$) prior to addition of HeLa whole-cell extracts (2 μg/μl) or FLAG-IP eluate. Following incubation, samples were treated with proteinase K extraction buffer (200 mM Tris–HCl pH 7.5, 25 mM EDTA pH 8.0, 300 mM NaCl, 2% SDS, 0.3 μg/μl proteinase K) at 65°C for 20 min; RNA was phenol–chloroform extracted, precipitated and loaded on 10% PAGE. Gels were exposed using phosphorimaging, and the obtained signal was quantified using the evaluation software ImageQuant (GE Life Sciences) and corrected by subtraction of appropriate background values. When relevant, recombinant archease (to a final

concentration of 0.2 µg/µl) or a corresponding volume of buffer (10 mM Tris–HCl pH 8.0, 100 mM NaCl, 1 mM dithiothreitol (DTT), 10% (v/v) glycerol) was added to the reaction. Decade marker (Ambion) was used for size determination for analytical PAGE.

### Protein and enzyme preparation for *in vitro* ligation assay

Recombinant IRE1 enzyme was obtained as described before (Volkmann *et al*, 2011). Cloning and preparation of recombinant hexahistidine-tagged human archease as well as generation of stably transfected HEK293 cell lines and affinity purification of FLAG-archease, FLAG-DDX1 and FLAG-RTCB was performed as described (Popow *et al*, 2014).

### Immunofluorescence

Wild-type HeLa cells were seeded on coverslips and treated with 300 nM thapsigargin (Tg) for 30 min, 4 or 16 h, respectively. Cells were fixed with 2% (w/v) paraformaldehyde at room temperature for 20 min, permeabilized by incubation in 0.2% (v/v) Triton X-100/PBS for 5 min and incubated in blocking solution (5% (w/v) BSA, 0.1% (v/v) Tween-20 in PBS, sterile-filtered) for 30 min at room temperature. Primary antibodies were dissolved in blocking solution and added to the coverslips for 1 h. The following antibodies and dilutions were used: RTCB (Santa Cruz, 1:500), archease (monoclonal, 1:2) and calnexin (Santa Cruz, 1:100). The archease monoclonal antibody was generated by immunizing mice with wild-type histidine-tagged archease purified as described previously (Popow *et al*, 2014). Cells were washed three times in 0.1% PBST and incubated with fluorescent secondary antibodies diluted in blocking solution for 1 h: Alexa Fluor 488 donkey anti-rabbit IgG (Invitrogen, 1:500), Alexa Fluor 568 donkey anti-mouse IgG (Invitrogen, 1:500), and Alexa Fluor 647 donkey anti-goat IgG (Invitrogen, 1:500). Coverslips were again washed four times with 0.1% PBST and subsequently mounted in ProLong Gold Antifade Mountant with DAPI (Invitrogen). Images were taken at least 24 h after mounting using a laser-scanning confocal microscope (LSM780, Zeiss).

### Electron microscopy

B220$^+$ B cells were stimulated for 3 days with LPS and then centrifuged for 3 min at 1,200 *g*. After centrifugation, the supernatant was carefully aspirated and cells were fixed using 2.5% glutaraldehyde in 1× PBS pH 7.4 for 1 h at room temperature. Cells were then rinsed with the same buffer, pelleted and resuspended in 3% low melting point agarose. Samples were post-fixed in 2% osmium tetroxide in ddH$_2$O, washed, dehydrated in a graded series of ethanol solutions and embedded in Agar 100 resin. Ultrathin sections were cut at a nominal thickness of 70 nm, post-stained with 2% aqueous uranyl acetate followed by Reynold's lead citrate. Sections were examined with an FEI Morgagni 268D (FEI, Eindhoven, The Netherlands) operated at 80 kV. Images were acquired using an 11 megapixel Morada CCD camera (Olympus-SIS).

### Subcellular fractionation

Wild-type HeLa cells were seeded at equal cell densities and treated with 300 nM Tg for 30 min or 4 h. Subcellular fractionation analysis

was performed using the Subcellular Protein Fractionation kit for Cultured Cells (Thermo Scientific) according to manufacturer's instructions. Equal percentages of the fractions obtained were subsequently used for Western blot analysis as described below.

### Western blotting

Tet-ON HeLa shRNA cell lines were treated with Dox for six consecutive days and stressed with 300 nM Tg over a 24-h time course. Cells were harvested after 0, 2, 4, 8, 16 and 24 h and lysed in high salt buffer (20 mM Tris pH 7.5, 400 mM NaCl, 0.5% (v/v) NP-40, 0.3% (v/v) Triton X-100) supplemented with 0.1 mM PMSF (Sigma) and protease inhibitor cocktail (Roche). Protein concentration was determined using Bradford assay (Bio-Rad), and 30 µg of each sample were separated by SDS–PAGE. For B cells, cells were counted, directly lysed in 1× SDS loading buffer at 95°C for 5 min and loaded at $1-2 \times 10^6$ cells per lane. Proteins were transferred to Immobilon-P membranes (Millipore) and probed with the following antibodies using standard protocols: lamin A/C (Sigma, 4C11 1:1,000), HSP90 (Abcam, 3A3 1:2,000), calnexin (Santa Cruz, C-20 1:1,000), RTCB (described previously (Popow *et al*, 2011), 1:5,000), monoclonal antibody against archease was generated by immunization of mice, fusion of splenocytes and generation of hybridoma (Monoclonal Antibody Facility, Max F. Perutz Laboratories, Vienna 1:500), β-actin (Sigma, A2006 1:5,000), XBP1s (BioLegend, 1:500), DDX1 (Bethyl, A300-521A 1:1,000), FAM98B (Sigma, HPA008320 1:500) and CGI-99 (Sigma, HPA039824 1:1,000).

### Cloning of shRNAs

For RNAi-mediated depletion of RTCB and/or archease, shRNAs were designed as described earlier (Dow *et al*, 2012). The respective 97-mer oligonucleotides (IDT, see sequences below; guide sequences are marked in bold) were cloned into the optimized miR-E backbone of RT3GEN (Fellmann *et al*, 2013) or into a derived construct expressing a blasticidin resistance cassette instead of Neo (RT3GEB). For cloning, the following primers were used (Fellmann *et al*, 2013): miR-E_fwd: 5′-TAC AAT ACT CGA GAA GGT ATA TTG CTG TTG ACA GTG AGC G-3′ and miR-E_rev: 5′-TTA GAT GAA TTC TAG CCC CTT GAA GTC CGA GGC AGT AGG CA-3′. A detailed description of the cloning procedure and the control shRNA used has been published elsewhere (Zuber *et al*, 2011).

Control shRNA 97mer: TGCTGTTGACAGTGAGCGCAGGAATTA TAATGCTTATCTATAGTGAAGCCACAGATGTATAGATAAGCATTAT AATTCCTATGCCTACTGCCTCGGA

RTCB shRNA 97mer: TGCTGTTGACAGTGAGCGACAGGTTGAA GGTGTTTTCTATTAGTGAAGCCACAGATGTA**ATAGAAAACACCTT CAACCTGC**TGCCTACTGCCTCGGA

Archease shRNA 97mer: TGCTGTTGACAGTGAGCGAAAGATGT TAGAGATTACAATTTAGTGAAGCCACAGATGTA**AATTGTAATCTCT AACATCTTC**TGCCTACTGCCTCGGA

### Generation of Tet-ON HeLa cell lines and Tet-RNAi studies

To generate ecotropically infectable Tet-ON HeLa cells, we constructed a lentivirus coexpressing the ecotropic receptor (EcoR), rtTA3 and Puro by shuttling the according expression cassette from pRIEP (Zuber *et al*, 2011) into the pWPXLd backbone (Addgene

plasmid 12258). HeLa cells transduced with pWPXLd-EF1-EcoR-IRES-rtTA3-PGK-Puro (pWPXLd-RIEP) were selected with 2 μg/ml puromycin (VWR) and subsequently transduced with ecotropically packaged RT3GEN Tet-shRNA expression vectors as described earlier (Zuber *et al*, 2011). For single knockdown conditions, cells were transduced with a single shRNA expression vector and selected with 1 mg/ml G418 (Gibco). Double knockdown cells were obtained by sequentially infecting two shRNA expression vectors and subsequent selection using 1 mg/ml G418 and 10 μg/ml blasticidin (VWR). Tet-regulated shRNA expression was induced by treatment of these cells with 1 μg/ml doxycycline (Dox, Sigma) added to the medium. Cell culture medium supplemented with selection antibiotics and Dox was replaced every second day.

### Quantitative reverse-transcriptase PCR

RNA from Tet-ON HeLa shRNA cell lines treated with Dox for six consecutive days and stressed with 300 nM Tg over a 24-h time course (harvesting after 0, 2, 4, 8, 16 and 24 h) was isolated using TRIzol reagent (Invitrogen). Total RNA was DNase-treated and reverse-transcribed using the Maxima First Strand cDNA synthesis kit for RT–qPCR with dsDNase (Thermo Scientific) according to the manufacturer's instructions. cDNA was diluted 1:10 before analysis by quantitative PCR using GoTaq qPCR Master Mix (Promega). The PCR was performed in a 20-μl reaction volume and pipetted using a Bravo LT96 Liquid Handling system (Agilent). The following exon–exon spanning primers were designed using Primer3 software (version 0.4.0): human *ACTB*: 5′-TTG CCG ACA GGA TGC AGA AGG A-3′ (fwd) and 5′-AGG TGG ACA GCG AGG CCA GGA T-3′ (rev); human *XBP1s*: 5′-GAG TCC GCA GCA GGT G-3′ [fwd, primer spanning the non-conventional exon–exon junction, reported previously (Majumder *et al*, 2012)] and 5′-GGA AGG GCA TTT GAA GAA CA-3′ (rev); human total *XBP1*: 5′-GCG CTG AGG AGG AAA CTG AAA AAC-3′ (fwd) and 5′-CCA AGC GCT GTC TTA ACT CC-3′ (rev); human *XBP1u*: 5′-ACT ACG TGC ACC TCT GCA G-3′ (fwd) and 5′-GGA AGG GCA TTT GAA GAA CA-3′ (rev); human *EDEM1*: 5′-GAT TCC ATA TCC TCG GGT GA-3′ (fwd) and 5′-ATC CCA AAT TCC ACC AGG AG-3′ (rev); human *DNAJB9*: 5′-TGC TGA AGC AAA ATT CAG AGA-3′ (fwd) and 5′-CCA CTA GTA AAA GCA CTG TGT CCA-3′ (rev); human *HSPA5*: 5′-GTG GAA TGA CCC GTC TGT G-3′ (fwd) and 5′-GTG GAA TGA CCC GTC TGT G-3′ (rev); human *CHOP*: 5′-CAT TGC CTT CTC CTT CGG G-3′ (fwd) and 5′-CCA GAG AAG CAG GGT CAA GA-3′ (rev); human *BLOS1*: 5′-GAG GCG AGA GGC TAT CAC TG-3′ (fwd) and 5′-GCC TGG TTG AAG TTC TCC AC-3′ (rev); human *PDGFRB*: 5′-GCT CAC ACT GAC CAA CCT CA-3′ (fwd) and 5′-TCT TCT CGT GCA GTG TCA CC-3′ (rev); mouse *Rtcb*: 5′-GTT TGC CAT AGG GAA CAT GG-3′ (fwd) and 5′-GGT TCT TAG CAA GCG GAC AC-3′ (rev); primers for mouse *Xbp1s* (Rodriguez *et al*, 2012), mouse total *Xbp1* (Iwakoshi *et al*, 2003a), mouse *Edem1* (Lisbona *et al*, 2009), mouse μS (Taubenheim *et al*, 2012), mouse μM (Taubenheim *et al*, 2012) and mouse *Actb* (Lisbona *et al*, 2009) have been described previously. The reaction was performed using the following parameters: 50°C for 10 min, 95°C for 5 min, followed by 60 cycles in total at 95°C for 10 s and 60°C for 30 s. The quality of PCR primers was evaluated by melting curve analysis, DNA gel electrophoresis of the PCR products and determination of amplification efficiency. The obtained data were analyzed according to the $\Delta\Delta C_t$ method normalizing to human

*ACTB* or mouse *Actb* mRNA levels. Additionally, expression levels were normalized to the untreated control sample.

### RT–PCR

Human *XBP1s* and *XBP1u* mRNA levels were monitored by semi-quantitative real-time PCR using cDNA synthesized from dsDNase-treated RNA as described above, RedTaq ReadyMix™ PCR Reaction Mix (Sigma) and the following primers: 5′- TAA TAC GAC TCA CTA TAG GGG AAT GAA GTG AGG CCA GT-3′ and 5′-AAT CCA TGG GGA GAT GTT CTG GAG-3′. For *ACTB* mRNA levels, the following primers were used: 5′-TTG CCG ACA GGA TGC AGA AGG A-3′ (fwd) and 5′-AGG TGG ACA GCG AGG CCA GGA T-3′ (rev). PCR products were resolved by agarose gel electrophoresis. Densitometric analysis was performed using Fiji software (version 1.47i) and corrected by subtraction of the appropriate background values.

### Northern blotting

Northern blot analysis was done as previously described (Karaca *et al*, 2014). Hybridization with 100 pmol of the following [5′-³²P]-labeled DNA probes was performed at 50°C overnight: leucine tRNA 5′ exon probe: 5′-CTT GAG TCT GGC GCC TTA GAC-3′; tyrosine tRNA 3′ exon probe 5′-TCG AAC CAG CGA CCT AAG GAT-3′; arginine tRNA 5′ exon probe: 5′-TAG AAG TCC AAT GCG CTA TCC-3′; isoleucine tRNA 5′ exon probe: 5′-TAT AAG TAC CGC GCG CTA ACC-3′; and methionine tRNA 5′ exon probe: 5′-GGG CCC AGC ACG CTT CCG CTG CGC CAC TCT GC-3′. Equal loading was confirmed by hybridization of the blots with a [5′-³²P]-labeled DNA probe detecting U6 snRNA (5′-GCA GGG GCC ATG CTA ATC TTC TCT GTA TCG-3′).

### Metabolic labeling

Tet-ON HeLa shRNA cell lines were treated with Dox for six consecutive days. B220⁺ B cells were stimulated with LPS for 4 days. Thereafter, cells were starved in DMEM without L-methionine and L-cysteine (Gibco) for 2 h at 37°C and subsequently cultured for 1 h at 37 °C in Met and Cys-free DMEM supplemented with 16 MBq/ml [³⁵S]-labeled methionine and cysteine (Perkin Elmer, EasyTag™ EXPRESS³⁵S Protein Labeling Mix). HeLa cells were lysed in lysis buffer (2% (w/v) SDS, 20 mM HEPES pH 7.4), and proteins were pelleted by acetone precipitation overnight. After spinning, the obtained protein pellet was resuspended in lysis buffer. Protein concentration was determined by BCA assay (Thermo Scientific). Scintillation counts were measured and normalized to the respective protein concentrations (HeLa cells) or cell numbers (B cells). For autoradiography, cells were directly lysed in 1× SDS loading buffer at 95°C for 5 min. Proteins were separated by SDS–PAGE and transferred to Immobilon-P membranes (Millipore). Following autoradiography, membranes were incubated with the indicated antibodies.

### Mice

The conditional *Rtcb*^tm1a(KOMP)Wtsi ES cells were purchased from the EUCOMM/KOMP-CSD collection (MGI-ID 4362526, clone G09). ES cells were injected into C57BL/6J-*Tyr*^c-2J blastocysts to generate *Rtcb*^fl-laczneo/+ mice. The *Rtcb*^fl/+ allele was obtained by crossing these mice to the FLPe (Rodriguez *et al*, 2000). To detect deletion of

the *Rtcb* allele, the following primers were used for PCR genotyping of *Rtcb* mutant mice: 5′-GCC AAG CAT GTC CTG TAG AC-3′, 5′-AGA AAA GGG ATG GCT GAG TC-3′ and 5′-GGT CCC TTT TGC CTT CTG-3′. The wild-type *Rtcb* allele was identified as a 1,320-bp, the *Rtcb*[fl] allele as a 1,488-bp, and the deleted allele as a 706-bp PCR fragment. Both *Rtcb*[fl/fl] and *Cd23*-Cre (Kwon *et al*, 2008) mice were maintained on the C56/Bl6 background. All animal experiments were done according to valid project licenses, which were approved and regularly controlled by the Austrian Veterinary Authorities.

### Immunizations and plasma cell analysis

Mice were injected intraperitoneally with 50 μg TNP-0.5-LPS (Bio-Search Technologies) in PBS. After 14 days, the frequencies of TNP-specific antibody-secreting cells in the spleen were determined using the mouse IgM ELISPOT Kit (Mabtech) according to the manufacturer's protocol. TNP-14-BSA (BioSearch Technologies)-coated plates were used for capturing total anti-TNP-IgM antibodies secreted by individual cells. After extensive washing, the spots were counted with an AID ELISPOT reader system (AID Diagnostika). The measurement of the spots was performed automatically using the Definiens Software Suite. Prior to the analysis, the image data were processed by performing a shading correction. The spots were found using threshold segmentation and subsequent watershed segmentation. The serum titer of TNP-specific IgM or IgM antibodies was determined by ELISA using plates that were coated with 10 μg/ml of TNP-14-BSA in PBS as described previously (Nutman, 2001) with slight modifications: goat anti-mouse IgM (μ-chain specific) peroxidase or biotinylated anti-kappa light chain/streptavidin-HRP (BioLegend) was used as secondary antibodies. SureBlue™ TMB Microwell Peroxidase Substrate (Kirkegaard & Perry Laboratories) was used as substrate. Reactions were stopped with TMB stop solution (Kirkegaard & Perry Laboratories), and absorbance was read at 450 nm on a Tecan Genios Pro Fluorescence, Absorbance Reader (Tecan Trading AG).

### *Ex vivo* B-cell stimulations

B cells were isolated from spleens by positive enrichment of B220[+] B cells by MACS sorting (Miltenyi Biotech). The purified B cells were plated at $0.3 \times 10^6$ cells/ml in IMDM medium (IMP/IMBA Media Kitchen) supplemented with 20% fetal calf serum (Sigma), 50 μM 2-mercaptoethanol, 3 mM glutamine (Sigma), 100 U/ml penicillin and 100 μg/ml streptomycin sulfate (Sigma), and 20 mM HEPES were subsequently treated for up to 4 days with 20 μg/ml LPS from *Escherichia coli* (Sigma). For cell proliferation analysis, the purified B cells were first stained with 5 μM CellTrace Violet reagent (Invitrogen) before stimulation according to the manufacturer's protocol. IgM present in cell culture supernatants was measured using the Mouse IgM ELISA Ready-SET-Go kit (eBioscience) according to the manufacturer's protocol. The numbers of IgM-specific ASCs in the cultures were determined using the mouse IgM ELISPOT Kit (Mabtech) according to the manufacturer's protocol.

### Flow cytometry

Mice at 6–8 weeks of age were used for FACS analysis of mature B-cell subsets. Single cell suspensions from spleen or *in vitro*

cultured B cells were incubated with CD16/CD32 Fc block (eBioscience) to inhibit unspecific antibody binding. For flow cytometry, cells were stained with the following antibodies: anti-B220 FITC (RA3-6B2), anti-IgM PE-Cy7 (II-41), anti-IgD FITC (11-26), anti-CD21 PE (8D9), anti-CD1d biot (1B1), CD4 APC (GK1.5), anti-CD8α APC (53-6.7) anti-CD8β APC (H35-17.2), anti-F4/80 APC (BM8), anti-CD28 PE-Cy7 (37.51) from eBioscience; anti-CD19 Pacific Blue (6D5), anti-CD23 Alexa Fluor 647 (B3B4), anti-B220 Brilliant Violet 780™ (RA3-6B2), anti-CD22 APC (Ox-27), streptavidin APC-Cy7 from BioLegend; and anti-CD138 PE (281-2) from BD. Flow cytometric analysis was performed using a LSRII instrument (BD Biosciences) and the FlowJo Software. For FACS sorting, the BD FACSAria or FACSAriaIII flow cytometer (BD Biosciences) was used.

### Statistical analysis

Results were statistically compared using a two-way ANOVA or an unpaired Student's *t*-test. A *P*-value of $P < 0.05$ was considered significant.

**Supplementary information** for this article is available online: http://emboj.embopress.org

### Acknowledgements

We are grateful to Jutta Dammann for technical support; Nicole Fellner and Harald Kotisch for electron microscopy; Marietta Weninger, Thomas Lendl and Gabriele Stengl for FACS sorting; Gerald Schmauss for FACS sorting and image analysis of ELISPOT data; Arabella Meixner, Aleksandra Smus and Esther Rauscher for help with ES cell work and mouse husbandry; Hans-Christian Theußl for blastocyst injection and generation of chimeric animals; Kazufumi Mochizuki for providing the HSP90 antibody; Tomas Aragon, Diego Rojas-Rivera, Claudio Hetz and Stefan Weitzer for advice; and all members of our laboratory for discussions and support. This work has been funded by the Fonds zur Förderung der wissenschaftlichen Forschung (FWF-P24687), the Institute of Molecular Biotechnology Austria (IMBA), a Hertha Firnberg PostDoctoral Fellowship (JJ), a Boehringer Ingelheim Fonds PhD Fellowship and the Doctoral Program for RNA Biology (TH) and the Young Investigator Program (YIP-EMBO) (JM).

### Author contributions

JJ and TH designed and carried out experiments and wrote the manuscript; AFN designed and carried out experiments; TK performed experiments; MM designed experiments; MB supervised the B cell part of the project; JP and KH contributed to experiments; TH designed the shRNA cloning strategy; JM supervised the project and contributed to writing the manuscript.

### Conflict of interest

AFN is employed at *The EMBO Journal* as a scientific editor. AFN was not involved in any way in the review process or the editorial evaluation of this manuscript and is not privy to the referee identities. The remaining authors declare that they have no conflict of interest.

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
