## [Review Process File · The EMBO Journal]

Manuscript EMBO-2014-90332

The mammalian tRNA ligase complex mediates splicing of XBP1 mRNA and controls antibody secretion in plasma cells

Jennifer Jurkin, Theresa Henkel, Anne Faerch Nielsen, Martina Minnich, Johannes Popow, Therese Kaufmann, Katrin Heindl, Thomas Hoffmann, Meinrad Busslinger and Javier Martinez

Corresponding author: Javier Martinez, IMBA

Review timeline:

Submission date:	17 September 2014
Editorial Decision:	30 September 2014
Revision received:	17 October 2014
Accepted:	17 October 2014

Editor: Karin Dumstrei

Transaction Report:

1st Editorial Decision

30 September 2014

Thank you for submitting your manuscript to The EMBO Journal. Your study has now been seen by two referees.

As you can see below, both referees find the analysis interesting and suitable for publication in The EMBO Journal. They also mention the recent Lu et al paper, but also find that your findings confirm and extend this analysis. The referees find the analysis well done and have no technical issues. There are only text revisions needed and the manuscript will not be re-reviewed. Referee #1 asks if you have data on LPS-mediated B cell activation prior to 4 days? If you have it, then please go ahead and include it. If not then respond to this issue either in the manuscript or in the point-by-point response.

I am keen on getting the revised manuscript back as soon as possible. Since there are just text changes needed I don't think it should take so much time to do so. Would it be possible to get the revised version back within a week?

REFeree REPORTS

Referee #1:

The paper by the group of Javier Martinez investigates the identity of the RNA ligase that joins together two fragments of Xbp1 mRNA to activate one of the branches of the unfolded protein response (UPR). It is well known that activated Ire1a cleaves XBP1u mRNA. In yeast the UPR was originally identified and Ire1, the HAC1 mRNA (xbp1 homolog) & the tRNA ligase, Rlg1, were all

characterized in the 1990s already.

The data presented in this study show convincingly that in mammalian cells XBP1 splicing depends on a subunit of the tRNA ligase complex, RTCB, acting in conjunction with archease. So, at long last the elusive Rlg1 homolog in humans has been found.

As expected, RTDB depletion jeopardizes differentiation of B lymphocytes into plasma cells, an event that requires the IRE1a/XBP1 pathway. The development of the ER and antibody secretion are severely affected in plasma cells lacking RTCB.

Altogether the results are convincing and clearly and concisely explained.

Even though the novelty is limited - a paper just appeared in Molecular Cell uncovering likewise the identity of the mammalian Xbp1 ligase -, confirmation in this manuscript is important. Moreover, it extends to a key hallmark of the UPR in physiology: B cell differentiation hinges on correct xbp1 splicing, and hence, ligase activity.

Considering the recent Mol Cell paper appearance publication should not be delayed. I would suggest therefore the authors only the following:

In the B cell differentiation field it is known that only later in the process do B cells start to produce antibody in bulk. Still at early time points after LPS activation the ER already expands (see van Anken et al. Immunity 2003). Little if any spliced XBP1 splicing is detected at these early stages, but the question is whether IRE1a driven XBP1 splicing nevertheless is crucial for the anticipatory ER expansion. The authors show the data for the LPS driven B cell activation at 4 days, but if they do have the data for earlier days they may wish to include those or otherwise discuss this issue.

A few typos should be corrected throughout the manuscript.

Referee #2:

This is a very important and conclusive work providing evidence that RtcB ligase is involved in splicing of XBP1 mRNA during UPR response in mammalian cells. The data are also provided that this is also the case in vivo, in mouse, during production of immunoglobulins in B cells. This is a logical and exciting extension of recent discoveries, reported by Martinez's lab, describing characterization of the tRNA splicing ligase and also archease, which has stimulatory effect on tRNA and also, as shown in the submitted manuscript, on XPB1 mRNA splicing. I have relatively minor issues, which should be addressed before publication:

1. The paper should be reformatted to comply with the EMBO J. format. This should also include expansion of Introduction and Discussion. The findings about the role of tRNA splicing in mammalian cells should be contrasted with analogous data for yeast where also tRNA splicing pathway is followed, but the mechanism of ligation is different than in mammals. In this context, since the findings report a novel role for the tRNA splicing ligase in mammals, one should refer in the Introduction to the papers originally describing the mammalian ligase, namely those by Laski et al, JBC1983 (Phil Sharp's group) and Filipowicz and Shatkin Cell 1983 and Filipowicz et al. NAR 1983.

2. Page 4. Line 5 top: "...through archease itself..." change this to "...depletion of archease...".

3. Page 4, line 6 top: is the reference to Fig. S1e correct? I find it confusing.

4. Page 4, line 2 bottom: "...only a minor...". The effect is clearly there though it does not reach statistical significance. Hence, I would rather say "...less strong reduction...". Also, when discussing this effect on page 5, line 4 top, the "almost unaffected" should be changed to "moderately affected" or so. Clearly effects in panels c and d contrast with Fig. S2e and f where indeed no effect is seen.

5. It would help to have Fig. 3d split into two panels, each containing 3 graphs.

6. Page 7, line 8 bottom: In Fig. S2a Ile and Arg tRNAs are used, and not Tyr and Leu as here, to illustrate the same point. Please comment why this different choice of tRNAs.

7. Page 8, upper paragraph.

Although not absolutely essential, it would be nice to have controls showing that also in the in vivo experiments, lasting longer, there is no deficiency of spliced tRNAs and defect in protein synthesis.

Alternatively, arguments against this possibility should be brought about.

8. The authors should also refer in Discussion to the related findings for ES cells recently published in Mol. Cell by Wang's group.

1st Revision - authors' response

17 October 2014

Referee #1:

We would like to thank Referee #1 for evaluating our manuscript and acknowledging our efforts. What follows is a point-by-point response to the Referee's concerns..

In the B cell differentiation field it is known that only later in the process do B cells start to produce antibody in bulk. Still at early time points after LPS activation the ER already expands (see van Anken et al. Immunity 2003). Little if any spliced XBP1 splicing is detected at these early stages, but the question is whether IRE1a driven XBP1 splicing nevertheless is crucial for the anticipatory ER expansion. The authors show the data for the LPS driven B cell activation at 4 days, but if they do have the data for earlier days they may wish to include those or otherwise discuss this issue.

In the study mentioned by the referee, Van Anken et al. reported an increased expression of ER proteins already one day after stimulating a B cell lymphoma cell line with LPS, which preceded the massive increase in IgM synthesis and XBP1s induction. In the *in vitro* system used in our study, B cells stimulated with LPS differentiate from activated B cells via pre-plasmablasts to plasmablasts. To accurately address the early kinetics of ER expansion and XBP1s induction during different stages of plasmablasts development (rather than time after LPS stimulation) one should fractionate distinct populations in bulk cultures. We have used the markers CD22 and CD138 to differentiate between these three B cell populations. However, we did not look at ER expansion or IRE1a/XBP1s induction at earlier stages of plasmablasts development. We agree, that it would be interesting to dissect the series of events during plasmablast development by analyzing distinct stages of B cell differentiation and to compare them to earlier studies performed on total B cell cultures or cell line models. Still, given our present knowledge, we would currently favor a model, in which activated B cells would initially induce differentiation into secreting cells and only upon increased production of secreted proteins induce splicing of *Xbp1* mRNA, which then would drive further ER expansion helping to sustain high antibody secretion.

A few typos should be corrected throughout the manuscript.

We have proofread the manuscript in detail in order to eliminate typos.

Referee #2:

We would like to thank Referee #2 for his suggestions and comments. What follows is a point-by-point response to Referee #2.

The paper should be reformatted to comply with the EMBO J. format. This should also include expansion of Introduction and Discussion. The findings about the role of tRNA splicing in mammalian cells should be contrasted with analogous data for yeast where also tRNA splicing pathway is followed, but the mechanism of ligation is different than in mammals. In this context, since the findings report a novel role for the tRNA splicing ligase in mammals, one should refer in the Introduction to the papers originally describing the mammalian ligase, namely those by Laski et al, JBC1983 (Phil Sharp's group) and Filipowicz and Shatkin Cell 1983 and Filipowicz et al. NAR 1983.

We reformatted the paper to comply with EMBO Journal and expanded the Introduction section. As part of our more comprehensive introduction we also compare the mechanism of tRNA ligation in mammalian cells to the yeast pathway, which employs a different enzyme to catalyze the same biochemical reaction (page 2, paragraph 2). Within the Introduction we also refer to the papers originally describing the mammalian ligase activity (page 2, paragraph 2).

Page 4. Line 5 top: "...through archease itself..." change this to "...depletion of archease...".

With this sentence we want to make clear that archease, in contrast to RTCB, does not exhibit any ligase activity. We admit that this was ambiguously phrased. We re-wrote the entire paragraph and hope that our claim is now expressed sufficiently clear (page 6, paragraph 1, lines 1-6).

Page 4, line 6 top: is the reference to Fig. S1e correct? I find it confusing.

We hope that by re-writing the paragraph mentioned in (2), the reference to Figure E1E is now appropriate.

Page 4, line 2 bottom: "...only a minor...". The effect is clearly there though it does not reach statistical significance. Hence, I would rather say "...less strong reduction...". Also, when discussing this effect on page 5, line 4 top, the "almost unaffected" should be changed to "moderately affected" or so. Clearly effects in panels c and d contrast with Fig. S2e and f where indeed no effect is seen.

We agree with the Referee. Although not significant, we can see moderate effects on the general stress responders CHOP and HSPA5, that are however less apparent than those seen on direct XBP1s downstream targets. In accordance with Referee #2's suggestion, we have therefore changed "...only a minor..." to "...less strong reduction..." (page 6, paragraph 3, line 7) and "...almost unaffected..." to "...moderately affected..." (page 7, paragraph 1, line 2).

It would help to have Fig. 3d split into two panels, each containing 3 graphs.

As proposed by the Referee, we have now split Figure 3D into two panels, each containing three graphs.

Page 7, line 8 bottom: In Fig. S2a Ile and Arg tRNAs are used, and not Tyr and Leu as here, to illustrate the same point. Please comment why this different choice of tRNAs.

Although tRNA introns occur only in a minor subset of tRNA genes, there is at least one tRNA isoacceptor family in each organism, in which all or almost all of the tRNA genes are encoded with introns. We have chosen Arg- and Ile-tRNA in the case of human, and Tyr- and Leu-tRNA in the case of mouse, based on the abundance of intron containing tRNA genes within particular tRNA families. Accordingly, mature tRNA levels of the here displayed isotypes should be most severely affected in the respective organism. We also comment now on our selection in the Results section (page 7, paragraph 2 line 3-7 and page 9, paragraph 3, line 10 to page 10, paragraph 1, line 1).

Page 8, upper paragraph. Although not absolutely essential, it would be nice to have controls showing that also in the *in vivo* experiments, lasting longer, there is no deficiency of spliced tRNAs and defect in protein synthesis. Alternatively, arguments against this possibility should be brought about.

We agree with Referee #2 in that, with the experiments done so far, we cannot rule out defects in mature tRNA levels or global protein translation rates in the *in vivo* immunization experiments that last substantially longer than the 4 day *ex vivo* stimulation protocol. Unfortunately both Northern blots and metabolic labeling experiments are complicated by the fact that even after immunization, plasma cells numbers in the spleen are extremely low (around 2×10^5 ; see Figure 5A). However, we have made several observations that argue against this possibility. Even in immortalized MEFs that have been depleted of RTCB for up to three weeks, we neither witnessed defects in global protein synthesis rates nor a drop of mature tRNA levels below 20% of wild type levels (unpublished data). However, concomitant with such drop in mature tRNA levels, we observed a growth arrest of these cells, that is caused by decreased proliferation rates rather than increased cell death (unpublished data). tRNAs have been reported to be extremely stable with half lives around weeks. Upon deletion of RTCB in differentiated cells that do not proliferate, we do not see significant changes in tRNA levels as compared to wild type cells (unpublished data). We thus propose that in resting cells, existing tRNA levels might be sufficient to sustain relatively normal protein synthesis rates and therefore assume that both *in vitro* and *in vivo* cell division numbers, rather than time after stimulation, might crucially influence the presence of mature tRNA and global protein synthesis in RTCB depleted B cells. We also comment on this issue in the Discussion section of the paper (page 12, paragraph 2).

The authors should also refer in Discussion to the related findings for ES cells recently published in Mol. Cell by Wang's group.

As proposed by Referee #2 we refer now in the discussion section to related findings that have recently been published by Lu et al. in Molecular Cell (page 11, paragraph 1, lines 2-6; page 11, paragraph 2, lines 10-11; page 12, paragraph 1, line 11-13, page 12 paragraph 2, lines 9-11).